# VideoGen: Generative Modeling of Videos using VQ-VAE and Transformers

## Abstract

We present VideoGen: a conceptually simple architecture for scaling likelihood based generative modeling to natural videos. VideoGen uses VQ-VAE that learns learns downsampled discrete latent representations of a video by employing 3D convolutions and axial self-attention. A simple GPT-like architecture is then used to autoregressively model the discrete latents using spatio-temporal position encodings. Despite the simplicity in formulation, ease of training and a light compute requirement, our architecture is able to generate samples competitive with state-of-the-art GAN models for video generation on the BAIR Robot dataset, and generate coherent action-conditioned samples based on experiences gathered from the ViZ-Doom simulator. We hope our proposed architecture serves as a reproducible reference for a minimalistic implementation of transformer based video generation models without requiring industry scale compute resources. Samples are available at https://sites.google.com/view/videogen.

## 1 Introduction

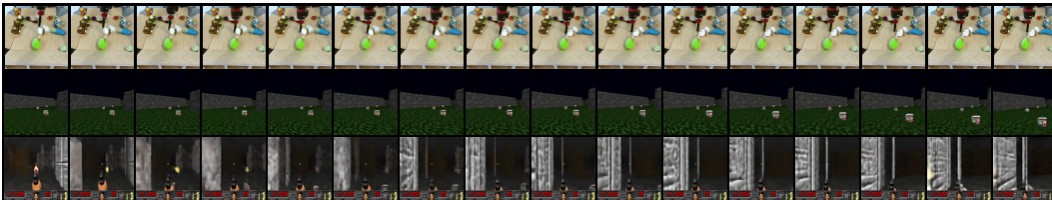

Figure 1: $64 \times 64$ samples for BAIR and ViZDoom environments generated by VideoGen

Deep generative models of multiple types (Goodfellow et al., 2014; van den Oord et al., 2016b; Dinh et al., 2016) have seen incredible progress in the last few years on multiple modalities including natural images (van den Oord et al., 2016c; Zhang et al., 2019; Brock et al., 2018; Kingma & Dhariwal, 2018; Ho et al., 2019a; Karras et al., 2017; 2019; Van Den Oord et al., 2017; Razavi et al., 2019; Vahdat & Kautz, 2020; Ho et al., 2020; Chen et al., 2020), audio waveforms conditioned on language features (van den Oord et al., 2016a; Oord et al., 2017; Bińkowski et al., 2019), natural language in the form of text (Radford et al., 2019; Brown et al., 2020), and music generation (Dhariwal et al., 2020). These results have been made possible thanks to fundamental advances in deep learning architectures (He et al., 2015; van den Oord et al., 2016b;c; Vaswani et al., 2017; Zhang et al., 2019; Menick & Kalchbrenner, 2018) as well as the availability of compute resources (Jouppi et al., 2017; Amodei & Hernandez, 2018) that are more powerful than a few years ago. However, one notable modality that has not seen the same level of progress in generative modeling is high fidelity natural videos. The complexity of natural videos requires modeling correlations across both space and time with much higher input dimensions, thereby presenting a natural next challenge for current deep generative models. The complexity of the problem also demands more compute resources which can be considered as one important reason for the slow progress in generative modeling of videos.

It is useful to build generative models of videos, both conditional and unconditional, as it implicitly solves the problem of video prediction and forecasting. Video prediction (Kalchbrenner et al., 2017;

Sønderby et al., 2020) can be seen as learning a generative model of future frames conditioned on the past frames. Architectures developed for video generation can be useful in forecasting applications for autonomous driving, such as predicting the future in more semantic and dense abstractions like segmentation masks (Luc et al., 2017). Finally, building generative models of the world around us is considered as one way to measure understanding of physical common sense (Lake et al., 2015).

Multiple classes of generative models have been shown to produce strikingly good samples such as autoregressive models (van den Oord et al., 2016b;c; Menick & Kalchbrenner, 2018; Radford et al., 2019; Chen et al., 2020), generative adversarial networks (GANs) (Goodfellow et al., 2014; Radford et al., 2015), variational autoencoders (VAEs) (Kingma & Welling, 2013; Kingma et al., 2016; Vahdat & Kautz, 2020), Flows (Dinh et al., 2014; 2016; Kingma & Dhariwal, 2018), vector quantized VAE (VQ-VAE) (Van Den Oord et al., 2017; Razavi et al., 2019), and lately diffusion and score matching models (Sohl-Dickstein et al., 2015; Song & Ermon, 2019; Ho et al., 2020). These different generative model families have their tradeoffs: sampling speed, sample diversity, sample quality, ease of training, compute requirements, and ease of evaluation.

To build a generative model for videos, we first make a choice between likelihood-based and adversarial models. Likelihood-based models are convenient to train since the objective is well understood, easy to optimize across a range of batch sizes, and easy to evaluate. Given that videos already present a hard modeling challenge due to the nature of the data, we believe likelihood-based models present fewer difficulties in the optimization and evaluation, hence allowing us to focus on the architecture modeling. Among likelihood-based models, autoregressive models that work on discrete data in particular have shown great success and have well established training recipes and modeling architectures.

Second, we consider the following question: Is it better to perform autoregressive modeling in a downsampled latent space without spatio-temporal redundancies compared to modeling at the atomic level of all pixels across space and time? Below, we present our reasons for choosing the former: Natural images and videos contain a lot of spatial and temporal redundancies and hence the reason we use image compression tools such as JPEG (Wallace, 1992) and video codecs such as MPEG (Le Gall, 1991) everyday. These redundancies can be removed by learning a denoised downsampled encoding of the high resolution inputs. For example, 4x downsampling across spatial and temporal dimensions results in 64x downsampled resolution so that the computation of powerful deep generative models is spent on these more fewer and useful bits. As shown in VQ-VAE (Van Den Oord et al., 2017), even a lossy decoder can transform the latents to generate sufficiently realistic samples. Furthermore, modeling in the latent space downsampled across space and time instead of the pixel space improves sampling speed and compute requirements due to reduced dimensionality. [1]

The above line of reasoning leads us to our proposed model: VideoGen, a simple video generation architecture that is a minimal adaptation of VQ-VAE and GPT architectures for videos. VideoGen employs 3D convolutions and transposed convolutions (Tran et al., 2015) along with axial attention (Clark et al., 2019; Ho et al., 2019b) for the autoencoder in VQ-VAE in order to be able to learn a downsampled set of discrete latents. These latents are then autoregressively generated by a GPT-like (Radford et al., 2019; Child et al., 2019; Chen et al., 2020) architecture. The latents are then decoded to videos of the original resolution using the decoder of the VQ-VAE.

Our results are highlighted below:

1. On the widely benchmarked BAIR Robot Pushing dataset (Ebert et al., 2017), VideoGen can generate realistic samples that are competitive with existing methods such as DVD-GAN (Clark et al., 2019), achieving an FVD of 112 when benchmarked with real samples, and an FVD* (Razavi et al., 2019) of 94 when benchmarked with reconstructions.
2. VideoGen can easily be adapted for action conditional video generation. We present qualitative results on the BAIR Robot Pushing dataset and Vizdoom simulator (Kempka et al., 2016).
3. We present ablations showing that employing axial attention blocks in the VQ-VAE and spatio-temporal position encodings in the Transformer are helpful design choices in VideoGen.
4. Our results are achievable with a maximum of 8 Quadro RTX 6000 GPUs (24 GB memory), significantly lower than the resources used in prior methods such as DVD-GAN (Clark et al., 2019) (32 to 512 16GB TPU (Jouppi et al., 2017) cores).

---

[1]Modeling long sequences is a challenge for transformer based architectures due to quadratic memory complexity of the attention matrix (Child et al., 2019).

## 2 BACKGROUND

### 2.1 VQ-VAE

The Vector Quantized Variational Autoencoder (VQ-VAE) (Van Den Oord et al., 2017) is a model that learns to compress high dimensional data points into a discretized latent space and reconstruct them. The encoder $E(x) \rightarrow h$ first encodes $x$ into a series of latent vectors $h$ which is then discretized by performing a nearest neighbors lookup in a codebook of embeddings $C = \{e_i\}_{i=1}^{K}$ of size $K$. The decoder $D(e) \rightarrow \hat{x}$ then learns to reconstruct $x$ from the quantized encodings. The VQ-VAE is trained using the following objective:

$$\mathcal{L} = \underbrace{\|x - D(e)\|_2^2}_{\mathcal{L}_{\text{recon}}} + \underbrace{\|sg[E(x)] - e\|_2^2}_{\mathcal{L}_{\text{codebook}}} + \underbrace{\beta \|sg[e] - E(x)\|_2^2}_{\mathcal{L}_{\text{commit}}}$$

where $sg$ refers to a stop-gradient. The objective consists of a reconstruction loss $\mathcal{L}_{\text{recon}}$, a codebook loss $\mathcal{L}_{\text{codebook}}$, and a commitment loss $\mathcal{L}_{\text{commit}}$. The reconstruction loss encourages the VQ-VAE to learn good representations to accurately reconstruct data samples. The codebook loss brings codebook embeddings closer to their corresponding encoder outputs, and the commitment loss is weighted by a hyperparameter $\beta$ and prevents the encoder outputs from fluctuating between different code vectors. An alternative replacement for the codebook loss described in Van Den Oord et al. (2017) is to use an EMA update which empirically shows faster training and convergence speed. In this paper, we use the EMA update when training the VQ-VAE.

### 2.2 GPT

GPT and Image-GPT (Chen et al., 2020) are a class of autoregressive transformers that have shown tremendous success in modelling discrete data such as natural language and high dimensional images. These models factorize the data distribution $p(x)$ according to $p(x) = \prod_{i=1}^{d} p(x_i|x_{<i})$ through masked self-attention mechanisms and are optimized through maximum likelihood. As in Vaswani et al. (2017), the architectures follow the standard design of employing multi-head self-attention blocks followed by pointwise MLP feedforward blocks.

## 3 VIDEOGEN

Our primary contribution is VideoGen, a new method to model complex video data in a computationally efficient manner. An overview of our method is shown in Fig 2.

**Learning Latent Codes** In order to learn a set of discrete latent codes, we first train a VQ-VAE on the video data. The encoder architecture consists of a series of 3D convolutions that downsample over space-time, followed by attention residual blocks. Each attention residual block is designed as shown in Fig 3, where we use LayerNorm (Ba et al., 2016), and axial attention layers follow Ho et al. (2019b).

The architecture for the decoder is the reverse of the encoder, with attention residual blocks followed by a series of 3D transposed convolution that upsample over space-time. The position encodings are learned spatio-temporal embeddings that are shared between all axial attention layers in the encoder and decoder.

**Learning a Prior** The second stage of our method is to learn a prior over the latents. The prior is learned by training a transformer model over the VQ-VAE latents. We follow the iGPT architecture with added dropout after the feedforward and attention block layers for regularization.

Although the VQ-VAE is trained unconditionally, we can generate conditional samples by training a conditional prior. We use two types of conditioning:

- **Concatenation**: We concatenate a conditional vector before every feedforward block in the transformer. This conditioning method is primarily used for frame conditioning, where the conditioned frame is encoded into a conditioning vector by a ResNet (He et al., 2016) backbone and then concatenated.

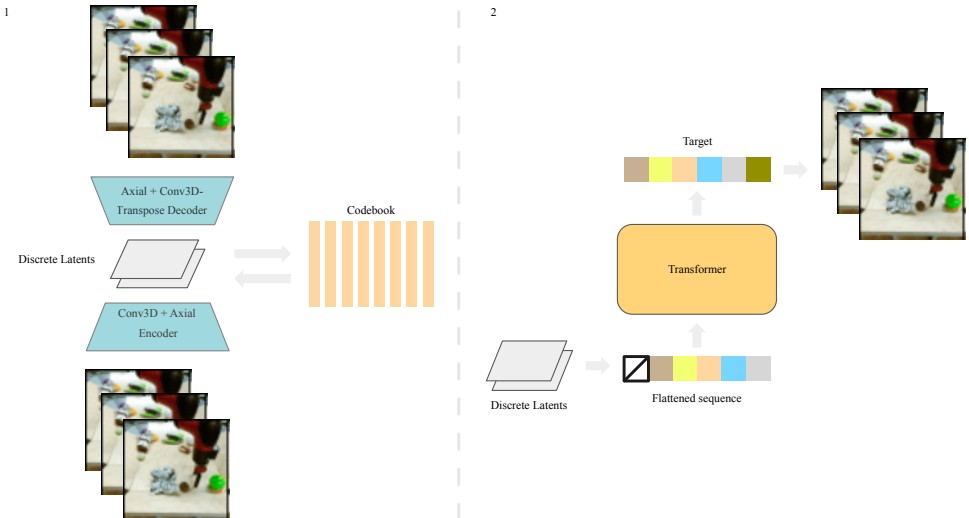

Figure 2: We break down the training pipeline into two sequential stages: training VQ-VAE (Left) and training a autoregressive transformer in the latent space (Right). The first stage is similar to the original VQ-VAE training procedure. During the second stage, VQ-VAE encodes video data to latent sequences as training data for the prior model. For inference, we first sample a latent sequence from the prior, and then use VQ-VAE to decode the latent sequence to a video sample.

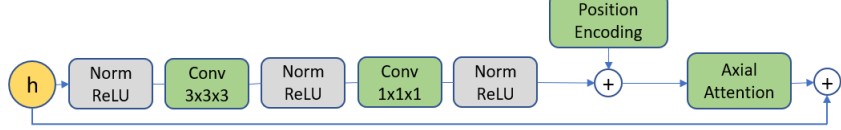

Figure 3: Architecture of the attention residual block in the VQ-VAE as a replacement for standard residual blocks.

- **Conditional Norms**: Similar to conditioning methods used in GANs, we parameterize the gain and bias in the transformer Layer Normalization (Ba et al., 2016) layers as affine functions of the conditional vector. This conditioning method is used for action conditioning.

## 4 EXPERIMENTS

In the following section, we evaluate our method and design experiments to answer the following questions:

- Can we generate high-fidelity samples from complex video datasets with limited compute?
- What architecture design choices in the VQ-VAE and transformer help the most?

### 4.1 TRAINING DETAILS

All image data is scaled to $[-0.5, 0.5]$ before training. For VQ-VAE training, we use random restarts for embeddings, and codebook initialization by copying encoder latents as described in Dhariwal et al. (2020). In addition, we found VQ-VAE training to be more stable (less codebook collapse) when using Normalized MSE for the reconstruction loss, where MSE loss is divided by the variance of the dataset. For all datasets, we train on $64 \times 64$ videos of sequence length 16. More training details can be found in Appendix A.

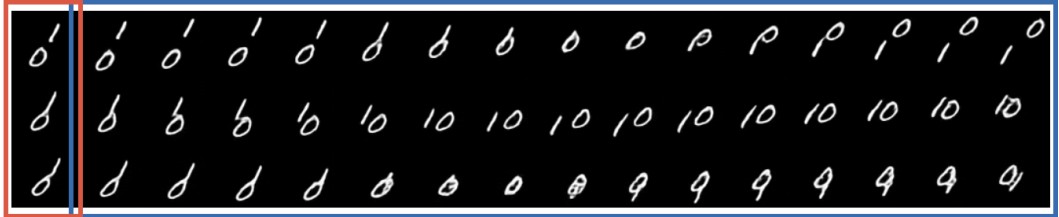

Figure 4: Moving MNIST samples conditioned on a single given frame (red).

## 4.2 MOVING MNIST

For Moving MNIST, the VQ-VAE downsamples by a factor of 4 over space-time (64x total reduction), and contains two residual layers with no attention. We use a codebook of 512 codes, each 64-dim embeddings. To learn the single-frame conditional prior, we train a conditional transformer with 384 hidden features, 4 heads, 8 layers, and a ResNet-18 single frame encoder. Fig 4 shows several different generated trajectories conditioned on a single frame.

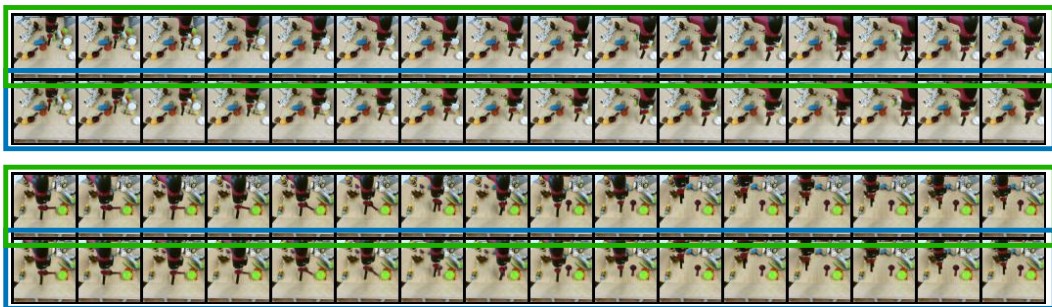

Figure 5: VQ-VAE reconstructions for BAIR Robot Pushing. The original videos are contained in green boxes and reconstructions in blue.

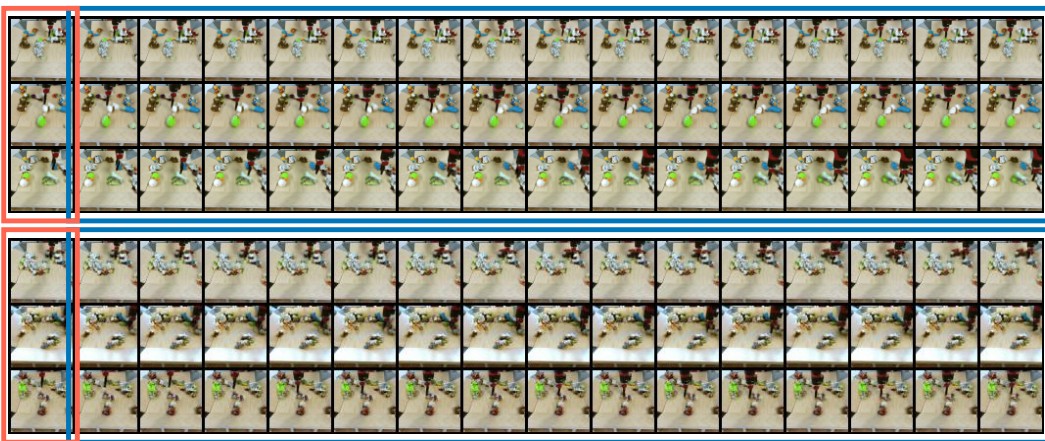

Figure 6: Samples for BAIR Robot Pushing. (Top) shows samples conditioned on a single frame. (Bottom) shows samples conditioned on a single frame and action sequence. Although scenarios are different in each trajectory, they all follow a similar action pattern. The conditional single frames are indicated by red boxes, and the sampled subsequent frames by blue.

Table 1: FVD on BAIR

| Method[2] | FVD ($\downarrow$) |
|---|---|
| SV2P | 262.5 |
| LVT | 125.8 |
| SAVP | 116.4 |
| **VideoGen (ours)** | 112.5 |
| DVD-GAN-FP | 109.8 |
| TrIVD-GAN-FP - | 103.3 |
| Video Transformer | $\mathbf{94 \pm 2}$ |

Table 2: Comparing FVD and FVD* values for BAIR on different transformer architectures. C = single-frame conditional, U = unconditional. The Axial Attention model follows the transformer architectures described in Ho et al. (2019b). GPT and GPT Small follow the same architecture, but GPT Small containing half the number of layers as GPT. FVD* is computed similar to FVD, but using reconstructed dataset examples instead of the original dataset examples

| | bits/dim | FVD | FVD* |
|---|---|---|---|
| Axial (C) | 3.94 | 170.1 | 133.3 |
| GPT Small (U) | 4.53 | 230.4 | 187.0 |
| GPT (U) | 4.08 | 191.7 | 164.2 |
| GPT (C) | **2.95** | **112.5** | **94.2** |

### 4.3 BAIR ROBOT PUSHING

For BAIR, the VQ-VAE downsamples by a factor of 4 over space, and 2 over time (32x reduction), with 4 attention residual blocks. We use a codebook of 2048 codes, each 256-dim embeddings. The transformer prior has a hidden size of 384 and 22 layers. For single-frame conditioning, we jointly train a ResNet-34 encoder that encodes the frame to a 512-dim vector.

Qualitatively, Fig 5 shows VQ-VAE reconstructions on BAIR. Fig 6 shows samples primed with a single frame as well as frame and action-conditioned samples. We can see that our method is able to generate good, realistic looking samples, and generalize to more out of distribution examples as shown in the case of samples for different priming frames with the same actions.

Quantitatively, Table 1[2] shows FVD results on BAIR, comparing our method with prior work. Although our method does not achieve state of the art, it is able to produce very realistic samples. To have a more thorough understanding of the model performance, we compute two extra metrics: (1) FVD of reconstructed VQ-VAE data examples, and (2) an adjusted FVD* metric similar to Razavi et al. (2019), where FVD is calculated between samples and a VQ-VAE reconstructed validation set (as opposed to the actual validation set). Table 2 shows that the adjusted FVD* of our single-frame conditional model is 94.2, coming very close the Video Transformer performance for FVD. This suggests that the FVD sample performance of our method is primarily bounded by the VQ-VAE reconstruction quality, as opposed to the transformer prior.

### 4.4 VIZDOOM

For ViZDoom, we use the same VQ-VAE and transformer architectures as for the BAIR dataset, with the exception that the transformer is trained without single-frame conditioning. We collect the training data by training a policy in each ViZDoom environment, and collecting rollouts of the final trained policies. The total dataset size consists of 1000 episodes of length 100 trajectories, split into an 8:1:1 train / validataion / test ratio. We experiment on the Health Gathering Supreme and Battle2 ViZDoom environments, training both unconditional and action-conditioned priors. Fig 7 and Fig 8 show samples from priors trained in each dataset and can see that the VQ-VAE and transformer are able to capture complex 3D camera movements and environment interactions. In addition, action-conditioned samples are visually consistent with the input action sequence and show a diverse range of backgrounds and scenarios under different random generations for the same set of actions.

---

[2]SV2P (Babaeizadeh et al., 2017), SAVP (Lee et al., 2018), DVD-GAN-FP (Clark et al., 2019), Video Transformer (Weissenborn et al., 2019), Latent Video Transformer (LVT) (Rakhimov et al., 2020), and TrIVD-GAN (Luc et al., 2020) are our baselines and we use FVD to compare different models given that adversarial models do not have a likelihood metric while the likelihood loss in VideoGen is on the discrete latents and not directly comparable to pixel level likelihood based models such as Video Transformer.

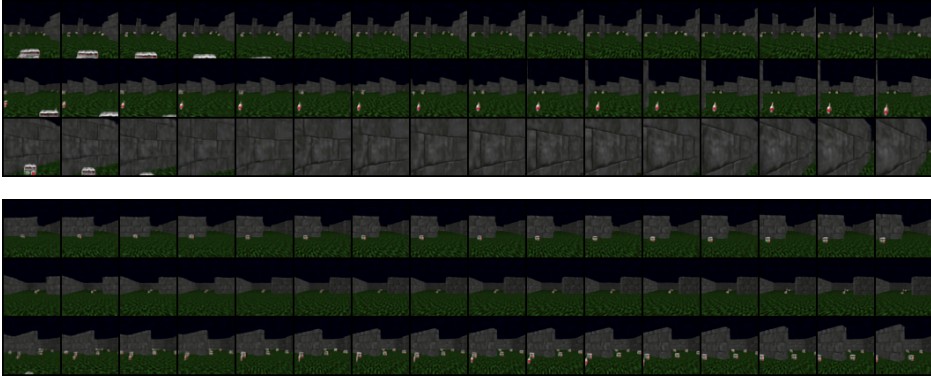

Figure 7: Samples for ViZDoom health gathering supreme environment. (Top) shows unconditionally generated samples. (Bottom) shows samples conditioned on the same action sequence (turn right and go straight).

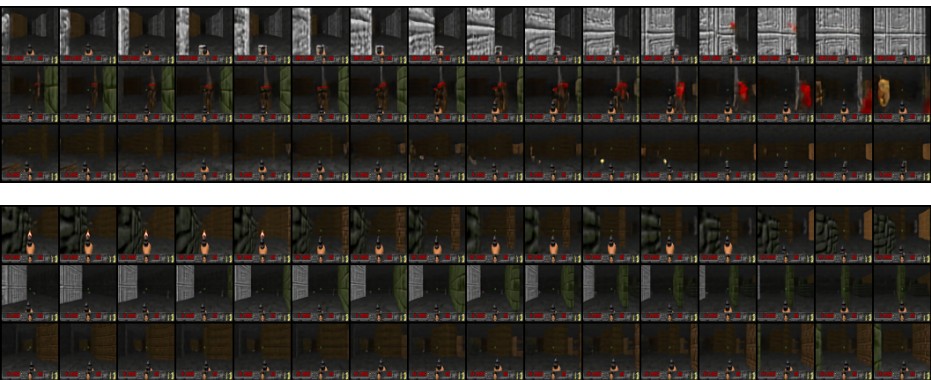

Figure 8: Samples for ViZDoom battle2 environment. (Top) shows unconditionally generated samples. (Bottom) shows three samples conditioned on the same action sequence (moving forward and right).

## 4.5 ABLATIONS

In this section, we perform ablations on various architectural design choice for VideoGen.

**Does attention in the VQ-VAE help?** We remove the axial attention layers from the VQ-VAE and compare with the original architecture as shown in Table 2. Empirically, incorporating axial attention into the VQ-VAE architecture improves reconstruction (NMSE) performance, and has much better reconstruction FVD compared to using no attention. Fig 5 shows a qualitative result of the VQ-VAE architecture with axial attention module.

**Are other forms of attention better for the transformer?** In addition to training a transformer with standard self-attention, we test our method using an Axial Transformer (Ho et al., 2019b) as a replacement. Table 3 shows an FVD comparison between a single-frame conditional Axial Transformer, and the standard GPT model using full self-attention. Using full attention achieves better FVD performance than an Axial Transformer. We hypothesize that this may be due to the full attention transformer being better able to capture single-frame conditional information than the Axial Transformer.

**Can a smaller transformer be used?** Computational efficiency is a primary advantage to our method, where we can first use the VQ-VAE to downsample by space-time before learning an autoregressive prior. Lower resolution latents allow us to train larger and more expressive prior to learn complex data distributions. In order to better understand why these advantages are important, we run an ablation on transformer size to demonstrate that training a larger transformer produces

Table 3: Ablation on attention in VQ-VAE. FVD is with reconstructed examples

| VQ-VAE Architecture | NMSE ($\downarrow$) | FVD ($\downarrow$) |
|---|---|---|
| No Attention | 0.014 | 156 |
| With Attention | **0.010** | **125** |

Table 4: Ablation on positional encodings

| Position Encoding | bits/dim ($\downarrow$) | FVD ($\downarrow$) |
|---|---|---|
| Sin-Cosine | 3.94 | 232 |
| None | 5.96 | 1697 |
| Temporal Only | 5.84 | 1360 |
| Spatial Only | 4.82 | 300 |
| Spatial + Temporal | **4.53** | **230** |

significantly better result. Table 2 shows the results of training a smaller unconditional transformer on BAIR, where it achieves both a worse bits per dim and FVD (230) compared to the larger transformer (192).

**Do learned spatiotemporal positional encodings help?** Finally, we study the effects of using learned spatiotemporal positional encodings when training the GPT model. Table 4 shows ablations on comparing different positional encodings. We see that adding in both space and time broadcasting aspects is crucial to better learn the data, and achieve a lower FVD. Using sine-cosine encodings performs roughly the same as spatiotemporal encodings, with a slightly worse FVD.

## 5 RELATED WORK

**Video Prediction** The problem of video prediction (Srivastava et al., 2015) is quite related to video generation in that the latter is one way to solve the former. Plenty of methods have been proposed for video prediction on the BAIR Robot dataset (Finn et al., 2016; Ebert et al., 2017; Babaeizadeh et al., 2017; Denton et al., 2017; Denton & Fergus, 2018; Lee et al., 2018) where the future frames are predicted given the past frame(s) and (or) action(s) of a robot arm moving across multiple objects thereby benchmarking the ability of video models to capture object-robot interaction, object permanance, robot arm motion, etc. Translating videos to videos is another paradigm to think about video prediction with a prominent example being `vid2vid` Wang et al. (2018). The `vid2vid` framework uses automatically generated supervision from more abstract information such as semantic segmentation (Luc et al., 2017) masks, keypoints, poses, edge detectors, etc to further condition the GAN based video translation setup.

**Video Generation** Most modern generative modeling architectures allow for easy adaptation of unconditional video generation to conditional versions through conditional batch-norm (Brock et al., 2018), concatenation (Salimans et al., 2017; van den Oord et al., 2016c), etc. Video Pixel Networks (Kalchbrenner et al., 2017) propose a convolutional LSTM based encoding of the past frames to be able to generate the next frame pixel by pixel autoregressively with a PixelCNN (van den Oord et al., 2016c) decoder. The architecture serves both as a video generative as well as predictive model, optimized through log-likelihood loss at the pixel level. Subscale Video Transformers (Weissenborn et al., 2019) extend the idea of Subscale Pixel Networks (Menick & Kalchbrenner, 2018) for video generation at the pixel level using the subscale ordering across space and time. However, the sampling time and compute requirements are large for these models. In the past, video specific architectures have been proposed for GAN based video generation with primitive results by Vondrick et al. (2016). Recently, DVD-GAN proposed by Clark et al. (2019) adopts a BigGAN like architecture for videos with disentangled (axial) non-local (Wang et al., 2017) blocks across space and time. They present a wide range of results, unconditional, past frame(s) conditional, and class conditional video generation. However, the DVD-GAN architecture is hard to replicate without industry scale compute resources. Furthermore, training DVD-GAN without a clean open source implementation is difficult due to instabilities often encountered in GAN training. Other examples of prior work with video generation of GANs include Saito et al. (2017), Tulyakov et al. (2018), Acharya et al. (2018), Yushchenko et al. (2019). In addition, Saito & Saito (2018) and Kahembwe & Ramamoorthy (2020) propose more scalable and efficient GAN models for training on less compute. Our approach builds on top of VQ-VAE (Van Den Oord et al., 2017) by adapting it for video generation. A clean architecture with VQ-VAE for video generation has not been presented yet and we hope VideoGen is useful from that standpoint. While VQ-VAE-2 (Razavi et al., 2019) proposes using multi-scale hierarchical latents, the pipeline is inherently more complicated. For simplicity, ease of reproduction and presenting the

first VQ-VAE based video generation model with minimal complexity, we stick with the single scale setup.

## 6 CONCLUSION

We have presented VideoGen, a new video generation architecture adapting VQ-VAE and Transformer models typically used for image generation to the domain of videos with minimal modifications. We have shown that VideoGen is able to synthesize videos that are competitive with state-of-the-art GAN based video generation models by requiring orders of magnitude fewer resources. We have also presented ablations on key design choices used in VideoGen which we hope is useful for future design of architectures in video generation. We hope that VideoGen serves as a simple baseline that is easy to reproduce and build upon for future research in this challenging topic.

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

# A  ARCHITECTURE DETAILS AND HYPERPARAMETERS

## A.1  VQ-VAE ENCODER AND DECODER

Table 5: Hyperparameters of VQ-VAE encoder and decoder models for Moving MNIST, ViZDoom (HGS = Health Gathering Supreme), and BAIR

|  | Moving MNIST | ViZDoom (HGS) | BAIR / ViZDoom (Battle2) |
|---|---|---|---|
| Input size | $16 \times 64 \times 64$ | $16 \times 64 \times 64$ | $16 \times 64 \times 64$ |
| Latent size | $4 \times 16 \times 16$ | $4 \times 16 \times 16$ | $8 \times 16 \times 16$ |
| $\beta$ (commitment loss coefficient) | 0.25 | 0.25 | 0.25 |
| Batch size | 64 | 64 | 64 |
| Learning rate | $7 \times 10^{-4}$ | $7 \times 10^{-4}$ | $7 \times 10^{-4}$ |
| Hidden units | 240 | 240 | 240 |
| Residual units | 128 | 128 | 128 |
| Residual layers | 2 | 4 | 4 |
| Uses attention | No | Yes | Yes |
| Codebook size | 512 | 2048 | 2048 |
| Codebook dimension | 64 | 256 | 256 |
| Encoder filter size | 3 | 3 | 3 |
| Upsampling conv filter size | 4 | 4 | 4 |
| Training steps | 20k | 75K | 75k |

## A.2  PRIOR NETWORKS

Table 6: Hyperparameters of prior networks for Moving MNIST, ViZDoom (HGS), BAIR and ViZDoom (Battle2).

|  | Moving MNIST | ViZDoom (HGS) | BAIR | ViZDoom (Battle2) |
|---|---|---|---|---|
| Input size | $4 \times 16 \times 16$ | $4 \times 16 \times 16$ | $8 \times 16 \times 16$ | $8 \times 16 \times 16$ |
| Conditional sizes | $1 \times 64 \times 64$ | 60 | $3 \times 64 \times 64, 64$ | 315 |
| Batch size | 32 | 32 | 32 | 32 |
| Learning rate | $3 \times 10^{-4}$ | $3 \times 10^{-4}$ | $3 \times 10^{-4}$ | $3 \times 10^{-4}$ |
| Vocabulary size | 512 | 2048 | 2048 | 2048 |
| Attention heads | 4 | 4 | 4 | 4 |
| Attention layers | 8 | 22 | 22 | 22 |
| Resnet depth | 18 | None | 34 | None |
| Resnet units | 512 | None | 512 | None |
| Dropout | 0.1 | 0.1 | 0.1 | 0.1 |
| Training steps 40k | 80k | 80k | 80k | 80k |

