# OpenReview forum: "VideoGen: Generative Modeling of Videos using VQ-VAE and Transformers"
_ICLR.cc/2021/Conference — Reject_

### Official Review · AnonReviewer4 · 2020-10-25
**Interesting, but lacks careful comparisons to the prior arts.**

**Rating:** 4
**Confidence:** 4

**Review:**

**[Quick summary ahead]**
Though I ended up with a negative score, I like the overall idea and intuition, as well as some of the analysis. However, without a clear distinction from prior arts, it is hard to call an interesting idea a contribution. Meanwhile, the analysis and ablation are less meaningful before other major problems fixed. Consequently, I do not list the pros of this work at this moment, since it is still unclear to me.

My main concerns are (corresponding to the 1-5 in the main review below):
    1. Requires a clear justification of the main contribution and tradeoffs, as well as a clear comparison with prior arts other than quality metrics (i.e., FVD) alone.
    2. The absence of a descriptive analysis regarding the performance or behavioral differences from prior arts.
    3. Writing problems in the methodology section.
    4. A better clarification of the main novelty.
    5. (Optional) Compare with a classical video generative model by modifying the architecture of CNNs and LSTM with VQ-VAE and GPT.


**[Main review]**
1. The paper lacks clarity in its positioning in terms of quality-efficiency tradeoff. In the experiment section, the quantitative results, as well as the authors' arguments (i.e., "Although our method does not achieve state of the art") agree that the performance is still not compatible with the state-of-the-art models. Meanwhile, in the abstract and conclusion, the authors mention: "our architecture is able to generate samples competitive with state-of-the-art GAN models." I am confused about such a situation. Could the authors make it less vague and consistent?
I understand this paper is not generation-quality-oriented. But when the method does not achieve state-of-the-art and claims it pose certain kinds of tradeoffs, then the authors are responsible for seriously quantifying and comparing the tradeoffs and show that the tradeoffs are significant and preferable.
Here, I list the tradeoffs claimed by the authors:
    - A. Simplicity in the formulation:
Not discussed in this paper. This is a very subjective statement but still needs to be carefully justified and compared against prior arts in the paper.
    - B. Ease of training:
Not discussed in this paper. What kind of ease? Training stability and convergence (then the authors should report a learning curve)? Hyperparameters agnostic (then the authors should report robustness against different hyperparameters)? Easy to reproduce (this point is awkward in all sense)? The authors should make it specific.
    - C. Light compute requirement:
        - (i) Only compared with DVD-GAN, which is well-known in training with super-large batch size (512), at the end of page 2. Do the authors try to evaluate the quality of DVD-GAN with the same amount of GPU resource (by reducing batch size and number of parameters) as VideoGen? Furthermore, optimally, the authors should compare with DVD-GAN using the same architectural design of VideoGen, as the architectures of VQ-VAE and GPT are not the main contribution of this paper.
        - (ii) The computational resource should be measured with GPU-days, not the number of GPUs. There are plenty of quick workarounds to reduce memory requirements, like reducing the batch size and the number of features. The real problem is whether the models can converge to a given performance with the same GPU-days.
        - (iii) What about the computational resource used by SAVP and Video Transformer listed in Table 1? Do VideoGen achieves a better tradeoff against those methods?
I believe the authors should take it more seriously on measuring the tradeoffs, especially when the tradeoffs are the main contributions.
2. To be honest, I am not very sensitive to the FVD score numbers (I know the definition, but not familiar with how large the perceptual differences are with given values). But the 146 FVD from VideoGen doesn't seem very close to its opponents with 116, 109 and 94 FVD. Especially the Video Transformer has only two points of variance, which may imply the degradation from 94 to 146 is quite a large number.
Furthermore, the background occupies a large area and is pretty static on the BAIR dataset. It is intuitively sound that, at least to me, such a performance difference is visually significant. Could the authors clarify the overall perceptual or behavioral differences between VideoGen from the other methods? I believe such a descriptive comparison and analysis are pretty common and should be presented in this paper.
3. The methodology section is very vague, nearly poorly written. The method is supposed to be the main contribution of this work, it is a bit hard to understand why this section is written carelessly.
    - A. "In order to learn a set of discrete latent codes." What is the shape and design of the latent code? Does it consider a temporal temporal dimension or just a flat code? The authors should specify the basic properties here, even if it is the same as what the authors have mentioned in the background section. The background is not a part of the proposed method.
    - B. "The prior is learned by training a transformer model over the VQ-VAE latents." I would not call this a clear explanation of how a model is applied. What are the inputs and outputs? How are the latents used as sequential data? The authors do provide Figure 2 to illustrate the idea, but Section 3 itself should be self-contained, and figures should be a complementary explanation of descriptive statements or mathematical forms. In fact, I believe the caption of Figure 2 should be presented in Section 3, but with more details.
    - C. "Conditional Norms." Where is the citation to this component? There are multiple different implementations of a conditional normalization layer. The authors even do not specify the type of normalization. It can be called a conditional batch normalization if it normalizes across a batch, or an adaptive instance normalization if it normalizes across channel dimensions within an instance.
4. Methodology-wise, the main contribution is more like partitioning a standard video generative model into two stages training, (a) reconstruction, and (b) diversity modeling. The VQ-VAE and GPT are only a change of backbone architecture design instead of the real contribution. I would recommend the authors rethink how they present their main contributions, instead of abusing the name of other well-known models.

5. (Continued 4.) Technically speaking, though the partitioning is reasonable, separating a jointly-optimizable model into a two-stage training pipeline is a bit awkward, and obviously responsible for a certain level of quality degradation.
I would recommend (not essential) the authors to have a baseline with the same architecture as VideoGen, but the VQ-VAE and GPT modules are jointly optimized (though expected to have a smaller batch size). I would be surprised if such a baseline does not perform better than VideoGen.


**[Minor comments]**

1. (Typo) Line 2-3 in the abstract, "learns learns"
2. The name of the model/method is too generic. It will be problematic for future papers referring to the proposed method. I would recommend the authors to make it more specific to the main feature or novelty of the proposed method.
3. Figure 2 is too sparse, while the texts in the figure are not friendly for reading.
4. The iGPT (I suppose is image-GPT), in Section 3 is never defined.

---

### Official Review · AnonReviewer1 · 2020-10-27
**Lack of clarity, evaluation, and novelty**

**Rating:** 4
**Confidence:** 4

**Review:**

After rebuttal:
Authors' responses do not address any of my concerns, and I completely agree with other reviewers regarding lack of clarity, evaluation, and novelty. The current form of the paper is not ready to be published. I decrease my score to reject.
--------------------------------------
Summary:
This paper presents a model combining VQ-VAE and GPT-like autoregressive model. Authors claim that the proposed model is light and easy to train compare to other generative models. The experiments on Moving MNIST, the BAIR Robot datasets, and ViZDoom show that the model can produce high quality and coherent action-conditioned samples.
--------------------------------------
Pros:
+ The model generates realistic and high quality video frames.
+ The proposed model is evaluated on diverse scenarios: the synthetic, robotics, and simulator-based dataset with action conditioned or unconditioned.
+ Section 4.5 ablations are very helpful.
--------------------------------------
Cons:
1. *Limited comparisons:*
    1. Quantitative comparisons on Moving MNIST and ViZDoom are missing.
    2. Qualitative results are shown on all datasets but without any comparisons with other models. Since this paper has limited quantitative comparisons and no qualitative comparisons, it is difficult to judge the performance of the model.
    3. I believe [1] is very related to the proposed model. Comparing with [1] is required.
2. *Justifying the claim about the light model:*
Authors claim that the proposed model requires light compute resources than other models. I understand the proposed model used significantly lower resources than others. The question is: is it due to less memory requirement or computations? To justify this claim, a comparison of model size, latency, and FLOPs with competing models, such as DVD-GAN-FP  [Clark2019], Video Transformer [Weissenborn2019], and Flow-based model [1] are necessary.
3. *Scaling-up with the proposed model:*
Authors claim that the proposed model is efficient due to autoregressive modeling on downsampled latent space. Other generative models like DVD-GAN-FP  [Clark2019] and Video Transformer [Weissenborn2019] provide an evaluation on Kinetics-600. Can the proposed model generate comparable or better video frames on more realistic datasets?
4. *Clarity:*
    1. In Figure 6 caption, authors claim that '(Bottom) shows samples conditioned on a single frame and action sequence. Although scenarios are different in each trajectory, they all follow a similar action pattern.' Does it mean that the samples are conditioned on the same action sequence for different video sequences? I don't think the examples are following a similar action pattern. It requires more explanations.
    2. Mistakes and typos.
        - I cannot find the result without the axial attention layers from the VQ-VAE in Table 2 (section 4.5 first paragraph).
        - The references of the models in Table 1 are missing.
        - Page 6 last sentence, 5 -> Fig 5.
    3. It would be easier to read the table 2 with more detailed caption.
        - a description of Axial
        - a difference between GPT and GPT Small
        - a description of bits/dim, the difference between FVD and FVD*
    4. Descriptions of experimental setup are missing. The results can not be easily reproduced.
        - Vizdoom data generation process
        - The number of samples in training/validation/test sets
        - The resolutions of videos
        - Any proprocessing (if there is any)
        - The architecture of GPT Small compare to GPT
5. *Missing references:* [1,2]
--------------------------------------
[1] Kumar, et al., VideoFlow: A Conditional Flow-Based Model for Stochastic Video Generation, ICLR 2020.
[2] Franceschi, et al., Stochastic Latent Residual Video Prediction, ICML 2020.

---

### Official Review · AnonReviewer3 · 2020-10-27
**Weakly evaluated, limited novelty and selective citation**

**Rating:** 4
**Confidence:** 5

**Review:**

Summary: Authors propose to model video by combining a VQ-VAE encoder-decoder model and a GPT model for the prior.

The primary contribution as stated by the authors: "Our primary contribution is VideoGen, a new method to model complex video data in a computationally efficient manner"
___________
Pros:
-
An interesting model and an ablation of its components.

___________
Cons:
-
- The primary contribution is stated but not validated. The claim is a new method to model complex video efficiently.
 - There is no experiments and/or benchmarks validating this claim anywhere in the paper.
 - There is work on efficiency in the video generation field that is neither cited nor benchmarked against.
   - TGANv2 (https://link.springer.com/article/10.1007%2Fs11263-020-01333-y) and LDVD-GAN (https://www.sciencedirect.com/science/article/abs/pii/S0893608020303397) come to mind.
   -  "Computational efficiency is a primary advantage to our method, where we can first use the VQ-VAE to downsample by space time before learning an autoregressive prior" - TGANv2, LDVD-GAN and DVD-GAN also do this .

- Some questionable highlights:
 - "VideoGen can generate realistic samples that are competitive with existing methods such as DVD-GAN"
   - A very weak highlight because several existing methods already do this better as shown in Table 1. and DVD-GAN is not the state-of-the-art for this benchmark as shown in the same table.
 - " VideoGen can easily be adapted for action conditional video generation"
   - This is applicable to every video generation model
 -  "Our results are achievable with a maximum of 8 Quadro RTX 6000 GPUs (24 GB memory),
significantly lower than the resources used in prior methods such as DVD-GAN"
   - This claim is not experimentally validated. DVD-GAN is also trainable on 8 Quadro RTX 6000 GPUs (24 GB memory). I would go further to argue that DVD-GAN would train faster and result in a higher performance than VideoGen. I would like to see a head to head benchmark or at the very least the wall clock time for training both the GPT prior and the VQ-VAE encoder-decoders.

- Selective Citation: The video generation and prediction field has been around for a long time now. It is hard to believe that the authors can manage to find and cite every relevant (un)published paper by google and deepmind authors yet they fail to find work published by other groups in this field. They then go on to talk about the slow progress in the field of video generation without acknowledging all the work being done in this field. The following statements highlight this:
 - "However, one notable modality that **has not seen the same level of progress** in generative modeling is high fidelity natural videos. "
 - " The complexity of the problem also demands more compute resources which can be considered as one important reason for the **slow progress** in generative modeling of videos."


- Missing References to published articles (related to the previous point)
   - TGAN: Temporal GAN - ICCV 2017 (First appeared on Arxiv - Nov 2016) - https://openaccess.thecvf.com/content_iccv_2017/html/Saito_Temporal_Generative_Adversarial_ICCV_2017_paper.html
   - MoCoGAN - CVPR 2018  (First appeared on Arxiv - Jul 2017) - https://openaccess.thecvf.com/content_cvpr_2018/html/Tulyakov_MoCoGAN_Decomposing_Motion_CVPR_2018_paper.html
   -  Progressive Video GAN - Masters Thesis (First appeared on Arxiv - Oct 2018) - https://arxiv.org/abs/1810.02419
   - MDP-GAN: Markov Decision Process for Video Generation - ICCV 2019 (First appeared on Arxiv - Sep 2019) - https://openaccess.thecvf.com/content_ICCVW_2019/html/HVU/Yushchenko_Markov_Decision_Process_for_Video_Generation_ICCVW_2019_paper.html
   - TGANv2: Train Sparsely, Generate Densely -  Journal of Computer Vision 2020 - (First appeared on Arxiv - Nov 2018) - https://link.springer.com/article/10.1007%2Fs11263-020-01333-y
   - LDVD-GAN: Lower Dimensional Kernels for Video Discriminators - Journal of Neural Networks 2020 -  (First appeared on Arxiv - Dec 2019) - https://www.sciencedirect.com/science/article/abs/pii/S0893608020303397
- If we were to include unpublished preprints on arxiv in this area, this list would at least double in size.



___________
Specific Points:
- "However, one notable modality that has not seen the same level of progress in generative modeling is high fidelity natural videos. The complexity of **natural videos** requires modeling correlations across both space and time with much higher input dimensions, thereby presenting a natural next challenge for current deep generative models"
 - The only natural video dataset benchmarked on is BAIR, the rest are all synthetic. Please benchmark on other datasets of natural video such as UCF101 and Kinetics-600 which also have comparative benchmarks at similar spatio-temporal resolutions.

- "Can we generate high-fidelity samples from complex video datasets with limited compute?"
   - Please address and expand on this point. It is currently left unanswered.

___________
Current recommendation: Rejection
-
All in all, this paper is lacking in novelty and does not do a good job of convincing readers of its primary contributions. The ablation studies provide for the most interesting insights with regard to this work. The BAIR evaluations show that the proposed model is more expensive and has a lower performance than many existing models. The claims of efficiency are also questionable given that the vqvae prior is notoriously expensive to train for image models, let alone video models and there is no head to head comparison or wall clock benchmark to demonstrate otherwise. Lastly, the very selective referencing of work situated around google and deepmind while ignoring related and highly relevant (and famous) work from scientists in other institutes is detrimental to research in this field. I am happy to update my review and score if these issues are addressed. But this work in it's current form is not publishable at any conference.

---

### Official Review · AnonReviewer2 · 2020-10-30
**Novelty seems to be incremental**

**Rating:** 4
**Confidence:** 5

**Review:**

This paper proposes a generative model to synthesize videos using VQ-VAEs. The scheme works in latent space by using embeddings for video sequences learnt by the VQ-VAE. For inference, an autoregressive transformer prior for video sequences is learnt, which upon sampling from and sending to the VQ-VAE decoder, generates unconditional (or conditional) samples of video. To learn video embeddings, the paper uses a 3D convolutional network, with an extra dimension for time.

Pros:
- Simple, principled setup
- Architectural novelties for videos (3D CNN, transformer prior)

Cons:
- I feel that the development is slightly incremental, compared with the original VQ-VAE work.
- Not enough analysis of latent space. For example, the original VQ-VAE work looks at a few experiments where the scene is traversed by moving 'forward' and 'right' (Figure 7 in [1]). I would have hoped that we had some experiments that show the virtues of working in latent space.
- Codebook collapse: it would be nice to have some more analysis of this component of the model.
- Other comments on analysis: There are many components in the setup, many of which need some discussion and analysis for this kind of work such as axial attention, the transformer model, latent spaces, etc.
- How does this model perform on larger image sequences, and larger number of timesteps?
- Other tasks: This work looks at the task of video generation. But there are many other areas of practical application where we can benefit from modeling video sequences. How does, for example, the model work with image segmentation, or tracking?

Overall: The work is interesting, but does not seem to have sufficient novelty other than having a different architecture design than used in the original VQ-VAE work. That being said, there's a lot to learn for practitioners if the authors were to put up a detailed write up on architectures and experiments.

[1] VQ-VAE: https://arxiv.org/abs/1711.00937

---

> ### Comment · AnonReviewer2 · 2020-11-25
> **Thank you for the response**
>
> I have read the author response and reviewer comments with interest. The work is definitely very interesting, but I keep my score because of the similar issues raised by reviewers - novelty, and evaluations, and computational aspects.

---

### Public Comment · ~Ruslan_Rakhimov1 · 2020-11-10
**The idea to use VQ-VAE+Transformer for video generation is not novel**

Applying transformer-based models in discrete latent space for video generation and prediction is not novel. The concurrent work and missing citation (Latent Video Transformer https://arxiv.org/abs/2006.10704) uses a combination of VQ-VAE+Video Transformer and beats VideoGen method on Bair Robot Pushing Dataset achieving 125.8 ± 2.9 FVD when benchmarked with real samples compared to 146 in your work. Also, they provide the available public code. As we can see, the most compelling case would be to see the results of this work on more challenging datasets like Kinetics 600. Last, your model uses 8 Quadro RTX 6000 GPUs (24 GB memory) for training, while LVT uses 8 V100 GPUs (16 GB memory). Could you also provide your model's inference time: how long does it take to generate a video?

---

> ### Author Response · Authors · 2020-11-20
> **Response**
>
> Thank you for responding. As you mentioned, it is "concurrent" work. we will definitely add a citation. we also note that we have much improved results than before and our FVD is now better than your work. We also plan to release our code in the near future

---

### Author Response · Authors · 2020-11-20
**Response to Reviewers**

We thank all reviewers for extensive feedback on our work. We believe it will be useful to improve our work and we have already uploaded a better version.

All reviewers raise the common concern that our work was not empirically rigorous in terms of the computation efficiency. We agree with this assessment and we will work on carefully comparing models with explicit hardware and FLOPs measures. However, we also would like to note that most papers in the generative modeling literature do not perform rigorous parameter/FLOP comparisons with respect to baselines. It has been well established that some of the best results in generative modeling have been achieved by labs with giant compute- infrastructure, packing as many model parameters and training as long as possible, using no fewer than 512 V100s [VQ-VAE-2, JukeBox, iGPT to name a few]. Work like ours that actually had the constraints to work with much less compute and ended up doing something sensible (training transformers on lower-resolution discrete latents as opposed to pixels) to address that, should be viewed as an example of ‘necessity is the mother of invention’ rather than being critically judged for lack of rigor that is generally absent among other papers in the field.

Second, regarding novelty, our paper never made any claim that we propose a novel architecture. Rather, we have been upfront that it is simply about combining what exists already: VQ-VAE and iGPT. Technically, no prior work has tried to show VQ-VAE can work on 3D data such as videos with pretty much no changes to the training pipeline except for architectural modifications like using 3D convolutions, axial attention, etc. We believe this finding by itself is important and useful to the community for future research on video generation using transformers and likelihood-based models. We see the minimum change from original VQ-VAE as a virtue rather than an issue. Our code and models will be made available for use by the community.

Third, with respect to selective citation: We were not aware of the work pointed out by the reviewer on video generation using GANs such as MoCoGAN, TGAN, etc. and we are happy to revise our draft, and add a discussion on these works. We however do think the tone of the reviewer could be more constructive than being combative and assuming we were selective about our citations. We would also like to point out that our quoted claims “video generation has not seen the same level of progress and one reason could be the amount of compute required” is true and is a widely accepted fact in the generative modeling community. In fact, this claim has been made in previous papers such as DVD-GAN.

---

> ### Comment · AnonReviewer4 · 2020-11-21
> **Response to Authors**
>
> > However, we also would like to note that most papers in the generative modeling literature do not perform rigorous parameter/FLOP comparisons with respect to baselines.
>
> It is because they don't claim computational-related novelties... For scientific writings, a claim must be followed with either a proof (experimental or theoretical) or a citation. But none of them showing up in the paper along with many claims on the contributions. I believe I should not need to emphasize this in a top-tier conference like ICLR, right?
>
> > It has been well established that some of the best results in generative modeling have been achieved by labs with giant compute- infrastructure, packing as many model parameters and training as long as possible, using no fewer than 512 V100s [VQ-VAE-2, JukeBox, iGPT to name a few].
>
> Do you mean ALL baselines use 512 V100s? It is unreasonable to make such kinds of statements that given only a few of those baselines use a large amount of computational resources. Actually, you can shrink down the computational resource of these baselines to that compatible with yours, as long as it is reasonable. We all understand that a superior performance can be achieved with over-parameterization and large-scale computation.
>
> Comparing to baselines is boring rather than showing your interesting findings. But it is crucial and one of the critical differences between a scientific publication and a technical blog/report.
>
> > Second, ...
>
> This work might be interesting for some people (I stay neutral here), but not ground-breaking and outweighing the problems in careless comparisons.
>
> (Then the non-neutral part) In fact, I can merely find interesting points in the paper if without computational contributions.
>
> > Third, ... We however do think the tone of the reviewer could be more constructive than being combative and assuming we were selective about our citations.
>
> 1. Does this mean that the authors do not believe missing citations is a critical problem and constructive feedback?
> 2. I do not see a particular review mention that the rejection is made due to the missing citations (except cases that the missed citation is a baseline that you should compare with). Most of the rejections are made with the lack of comparisons.

---

> > ### Author Response · Authors · 2020-11-21
> > **Response to Reviewer Response**
> >
> > >It is because they don't claim computational-related novelties... For scientific writings, a claim must be followed with either a proof (experimental or theoretical) or a citation. But none of them showing up in the paper along with many claims on the contributions. I believe I should not need to emphasize this in a top-tier conference like ICLR, right?
> >
> > We would like to point out that we never claim "novelty" anywhere. We have already mentioned this in our first response, but for the sake of completeness, our method is not inventing anything new nor does it have any novelty to improve the computational requirements of generative models.
> >
> > 1. It is well known that training an autoregressive model on discrete latents of a VQ-VAE is less expensive than training an autoregressive model on the pixels [VQ-VAE, VQ-VAE-2, Jukebox]. Why? Because the VQ-VAE performs spatial (and temporal) downsampling of the original resolution. For example, if you had a stride of 4 across Height, Width, and Time, you get to model 64x fewer tokens. The computation and memory of the Transformer (for autoregressive modeling) scales quadratic with the sequence length. 64x reduction => 4096x reduction for training the Transformer.
> >
> > 2. Have we invented this principle? No. Neither do we claim that. This is an idea put forth in the VQ-VAE paper by Van Den Oord et al (2017), and we merely adopt it for videos where we believe the need for such a downsampling is more given the larger resolutions of the inputs.
> >
> > 3. Computational Tradeoffs with respect to GANs: We agree with the reviewer that our claims on this front were loose. By which we mean that we were not rigorous about it. To say DVD-GAN required 512 GPUs while our method required 8 GPUs without a careful calibration of FLOPs, parameters, wall clock time is indeed not rigorous. We are happy to remove the computational-tradeoff statements comparing to DVD-GAN (or GANs in general) in the paper.
> >
> > 4. What does our proposed approach still have to offer when compared to GANs?: DVD-GAN public implementation by authors from DeepMind was never released. It is heavily based on top of the BigGAN style architecture and it is known that such models require large batch sizes for working well. Therefore, even if the models were smaller or consumed less FLOPs, it is likely that it maybe hard to train them with small batch sizes due to the instability typically present in GAN training. We believe our proposed approach is extremely simple, only uses well established tools such as vector-quantization, autoencoders and GPT,  all of which are easy to reproduce and run and are very much robust to using small batch sizes due to the nature of the loss (log-likelihood). We will release a public codebase very soon so that it is useful for the community.
> >
> > >Does this mean that the authors do not believe missing citations is a critical problem and constructive feedback?
> >
> > No. Missing citations is a critical problem and is a "useful" feedback. We have also "acted" on it by updating our related work section. However, the nature in which it was pointed out was not constructive. The reviewer clearly assumes we were selective about our citations rather than pointing out we missed citing paper X, Y, Z and why it is important to cite them.
> >
> > >I do not see a particular review mention that the rejection is made due to the missing citations (except cases that the missed citation is a baseline that you should compare with). Most of the rejections are made with the lack of comparisons.
> >
> > We do not say that either.
> >
> > >This work might be interesting for some people (I stay neutral here), but not ground-breaking and outweighing the problems in careless comparisons.
> >
> > Not groundbreaking indeed, but neither do we think it is a necessary condition for a paper to be useful to people in the community.
> >
> > >Comparing to baselines is boring rather than showing your interesting findings. But it is crucial and one of the critical differences between a scientific publication and a technical blog/report.
> >
> > Considering that we agree a lot on this and also that our approach is meant for a different class of generative models (and not GANs), could you clarify what comparisons and revisions you would specifically like to see?

---

> > > ### Comment · AnonReviewer4 · 2020-11-22
> > > **Response**
> > >
> > > > We would like to point out that we never claim "novelty" anywhere.
> > >
> > > Quoting from the list of contributions in the introduction, which shall mean the highlight of the paper,
> > > > > Our results are achievable with a maximum of 8 Quadro RTX 6000 GPUs (24 GB memory), significantly lower than the resources used in prior methods such as DVD-GAN (Clark et al., 2019) (32 to 512 16GB TPU (Jouppi et al., 2017) cores).
> > >
> > > It is clearly a claim on computational contribution. It is also the main reason why this work does not necessarily need to outperform DVD-GAN. Meanwhile, in Table 1 (in the initial submission), there are plenty of other baselines that perform better than the proposed method. It is straightforward to request the authors to provide similar but more rigorous statements and experiments to show that the performance is not due to the different levels of computational resources.
> > >
> > > > DVD-GAN public implementation by authors from DeepMind was never released.
> > >
> > > This is not even an excuse, any reviewer or advisor will say, "provide a self-implementation with the best of your effort in this case."
> > >
> > > > It is heavily based on top of the BigGAN style architecture and it is known that such models require large batch sizes for working well.
> > >
> > > A larger batch size makes it better (the main thing BigGAN shows), but it is not a requirement. The first version of the BigGAN architecture is identical to the Self-Attention GAN.
> > >
> > > > Not groundbreaking indeed, but neither do we think it is a necessary condition for a paper to be useful to people in the community.
> > >
> > > Well, then there are not too many things to discuss.
> > >
> > > > could you clarify what comparisons and revisions you would specifically like to see?
> > >
> > > Shall I copy and paste the initial review here? A good side of OpenReview is that people can see and learn from how the regular rebuttal undergoes. I would recommend the authors look around and see how other authors respond to the reviews.

---

### Decision · Program_Chairs · 2021-01-07
**Final Decision**

**Decision:**

Reject

**Comment:**

The paper focuses on the problem of high quality video generation. It approaches the problem by extending VQ-VAE to videos, where a GPT is used to model the low dimensional representation of the VAE. As agreed upon by the authors and the reviewers, the proposed method is simple and produces interesting results.

Based on all the reviews and the subsequent discussions, it seems that the reviewers' comments were mostly not addressed and they maintain their stance with regards to the paper's technical novelty, empirical justification of the paper's claims (specifically the claim on computational efficiency), and the rigorous comparison with prior art. The authors themselves make it clear that technical novelty was not the main driving force in this paper. However, in this case, it would be expected that the major claims of the paper be very clearly justified (especially with experiments and analysis) and comparison with other methods be more thorough. It seems that these latter two points remain in the latest revision of this paper. Since the paper shows promise, the authors are recommended to take the reviewers' comments and suggestions into consideration to produce a stronger and more thorough submission in the future.